# Bright middle cerebellar peduncle sign in multiple system atrophy with predominant cerebellar ataxia is more apparent in double-inversion recovery imaging than in conventional imaging

**Wataru Shiraishi**[1,2]*, **Ayano Matsuyoshi**[1,3], **Yusuke Nakazawa**[1], **Yukiko Inamori**[1], **Akira Yogi**[4], **Tetsuya Hashimoto**[1,5]

1 Department of Neurology, Kokura Memorial Hospital, Kitakyushu City, Fukuoka, Japan, 2 Shiraishi Internal Medicine Clinic, Nogata City, Fukuoka, Japan, 3 Department of Radiology, Graduate School of Medical Science, University of the Ryukyus, Nishihara Town, Okinawa, Japan, 4 Department of Neurology, Neurological Institute, Graduate School of Medical Sciences, Kyushu University, Fukuoka, Japan, 5 Department of Comprehensive Strokology, Fujita Health University School of Medicine, Toyoake, Aichi, Japan

* whitestone_db5@yahoo.co.jp

**Data Availability Statement:** Information on this paper can be obtained by contacting the corresponding author or Department in charge of

## Abstract

Multiple system atrophy (MSA) is a neurodegenerative disorder that presents as parkinsonism, cerebellar ataxia, and autonomic dysfunction. Magnetic resonance imaging (MRI) findings of MSA are reported to be the atrophy of the putamen/pons/cerebellum, hot cross bun sign, and bright middle cerebellar peduncle (MCP) sign. Here, we assessed the sensitivity of detecting the bright MCP sign in patients with MSA cerebellar variant (MSA-C) using a double inversion recovery (DIR) procedure, comparing it to the sensitivity of detection by T2-weighted image (T2WI) and fluid-attenuated inversion recovery (FLAIR) sequences on conventional MRI. We evaluated 6 MSA-C patients and 6 control patients (multiple sclerosis, neuromyelitis optica, and spinocerebellar atrophy). Characteristics of all the patients were collected, and MRI was analyzed. Two neurologists independently evaluated the visualization of the bright MCP sign using a 4-point visual grade from Grade 0 to Grade 3. Differences in grade between DIR and T2WI or FLAIR were statistically analyzed. Also, as a quantitative analysis, the signal intensity of the MCP lesion was compared with that of the ipsilateral thalamus, and the MCP/thalamus ratio was evaluated. As a result, DIR more sensitively showed the bright MCP signs of MSA-C patients than T2WI or FLAIR. Also, the bright MCP sign deteriorated and expanded over time in the cases we followed with MRI. We also evaluated hot cross bun sign in the pons, but the hot cross bun sign in MSA-C patients was not significantly different between the DIR and conventional MRI sequences. The DIR procedure can be a more sensitive method for detecting the involvement of MCP lesions in MSA-C.

Kokura Memorial Hospital (tel: +81-93-511-2000, email: rinsyo@kokurakinen.or.jp, URL: http://www.kokurakinen.or.jp/bumon/rinsyoukenkyu/ippan/) only for the parts of the information that do not conflict with private information.

**Funding:** The author(s) received no specific funding for this work.

**Competing interests:** The authors have declared that no competing interests exist.

## Introduction

Multiple system atrophy (MSA) is a neurodegenerative disorder characterized by parkinsonism, cerebellar ataxia, and autonomic dysfunction [1]. MSA is differentiated into two groups: MSA-cerebellar variant (MSA-C), characterized by a predominance of cerebellar ataxia, and MSA-parkinsonian variant (MSA-P), characterized by a predominance of parkinsonism. The pathological finding of MSA is oligodendroglial cytoplasmic inclusions (GCIs) composed of α-synuclein [2]. In MSA-C, GCI accumulation and demyelination occur mainly in the cerebellum and pontine structure, and the degree of GCI accumulation and demyelination increases with disease progression [3]. The magnetic resonance imaging (MRI) findings of structural alterations in MSA-C reflect these pathological changes. Gilman's criteria include atrophy of the pons and middle cerebellar peduncle (MCP) as imaging features of MSA-C [1]. Also, other MRI signs, such as the hot cross bun sign of the pons and bright MCP sign (hyperintensities in MCP on T2-weighted image [T2WI] or fluid-attenuated inversion recovery [FLAIR] sequences), have been described as additional features of MSA-C [4]. As a cross-shaped hyperintensity in the pons on T2WI, the hot cross bun sign has been reported to be a typical feature of MSA-C [5]. The bright MCP sign has also been reported to be helpful in the diagnosis of MSA-C [6], and the frequency of MCP abnormality in MSA-C cases is reported to be over 90% [7]. Also, the presence of the bright MCP sign is reported to increase the specificity but lower the sensitivity of MSA-C diagnosis [8]. The MCP abnormalities on MRI are reported to represent myelin loss [9], so the bright MCP sign of MSA-C exhibits a high T2WI signal intensity. However, the infratentorial lesion is surrounded by cerebrospinal fluid (CSF) and has a white matter-rich construction, which makes it difficult to identify the bright MCP signs by T2WI. A double inversion recovery (DIR) pulse sequence provides two different inversion pulses, which attenuate the CSF and white matter signal strength, thus achieving a superior delineation between gray and white matter [10]. The previous report showed that DIR is superior in detecting infratentorial lesions in multiple sclerosis patients [11]. Therefore, we hypothesized that DIR might be helpful in detecting bright MCP signs in MSA-C patients.

This study aimed to investigate whether DIR was significantly better than T2WI or FLAIR at detecting the bright MCP sign in MSA-C patients. Furthermore, in cases where we could follow the MRI changes over time, we evaluated the time course of the change in the MCP abnormality.

## Materials and methods

### Study population

We conducted a retrospective study on consecutive patients who underwent brain MRI including DIR between April 2021 and March 2022. MRI evaluations were performed for 6 consecutive MSA-C patients (3 men, 3 women; mean age, 60.7 years; age range, 48–83 years) who had been clinically diagnosed with probable or possible MSA-C according to the Consensus Criteria [1]. As a control group, patients with other neurological disorders (3 with multiple sclerosis, 2 with neuromyelitis optica, and 1 with spinocerebellar atrophy) were analyzed (Tables 1 and 2).

Table 1 lists patients baseline information. At the time of evaluation, all MSA patients had ataxia and autonomic dysfunction, and two patients showed parkinsonism. The disease duration was less than 5 years, and the time from onset to MRI imaging ranged from 6 months to 2 years and 3 months. At our hospital, DIR is not included in the standard MRI protocol, and it is performed when the physician determines that DIR is necessary. Therefore, there are no patients with common Parkinson's disease in this study. For this reason, the control group in

**Table 1. Characteristics of the patients with multiple system atrophy cerebellar variant (MSA-C) and the control patients.**

| Case | age (year) | sex | disease | ataxia | uni/bilateral symptoms | uni/bilateral MCP sign | autonomic dysfunction | parkinsonism | family history | duration (year) | onset to MRI | MCP signal grading (DIR) | MCP/Th ratio (DIR) |
|---|---|---|---|---|---|---|---|---|---|---|---|---|---|
| 1 | 36 | F | MS | (-) | n.a. | n.a. | (-) | (-) | (-) | 2 | 3m | 0 | 1.34 |
| 2 | 44 | M | MS | (-) | n.a. | n.a. | (-) | (-) | (-) | 1 | 1m | 0 | 0.92 |
| 3 | 45 | M | MS | (+) | n.a. | n.a. | (+) | (-) | (-) | 3 | 1y4m | 3 | 1.71 |
| 4 | 53 | M | NMO | (-) | n.a. | n.a. | (+) | (+) | (-) | 2 | 1y2m | 1 | 1.06 |
| 5 | 46 | F | NMO | (-) | n.a. | n.a. | (-) | (-) | (-) | 10 | 8y | 0 | 1 |
| 6 | 65 | F | SCD | (+) | n.a. | n.a. | (-) | (-) | (+) | 10 | 9y | 0.5 | 0.46 |
| 7 | 43 | M | MSA-C | (+) | bilateral | bilaterla | (+) | (-) | (-) | 3 | 1y4m | 3 | 3.56 |
| 8 | 62 | F | MSA-C | (+) | bilateral | bilateral | (+) | (-) | (-) | 2 | 6m | 2 | 1.65 |
| 9 | 63 | M | MSA-C | (+) | uniateral | unilateral | (+) | (-) | (-) | 3 | 1y | 3 | 2.91 |
| 10 | 65 | M | MSA-C | (+) | bilateral | bilateral | (+) | (-) | (-) | 3 | 1y4m | 3 | 1.87 |
| 11 | 83 | F | MSA-C | (+) | bilateral | unilateral | (+) | (+) | (-) | 4 | 2y3m | 2.5 | 1.74 |
| 12 | 48 | F | MSA-C | (+) | bilateral | bilateral | (+) | (+) | (-) | 1 | 6m | 3 | 3.12 |

For the control group, double inversion recovery (DIR) was obtained early in the onset of the disease (multiple sclerosis). In the cases of neuromyelitis optica, the course was longer and additional DIRs were performed over many years of follow-up. The association between disease duration and MCP signal intensity in MSA-C patients was insignificant.

DIR: double inversion recovery, F: female, M: male, m: month, MCP: middle cerebellar peduncle, MSA-C: multiple system atrophy cerebellar variant, MRI: magnetic resonance imaging, MS: multiple sclerosis, n.a.: not assessed, NMO: neuromyelitis optica, SCA: spinocerebellar atrophy, Th: thalamus, y: year.

this study is mostly patients with demyelinating diseases. No previous reports have focused on brainstem structure with DIR, and it was not known how the MCP and other brainstem structures would appear in healthy subjects or patients with other diseases. Therefore, we used patients with other diseases as a control group (reference group). The study protocol was approved by the Institutional Review Board of Kokura Memorial Hospital (21102001). The requirement for written informed consent was waived due to the study design's retrospective nature. This study protocol complied with Declaration of Helsinki principles. We last accessed these data on June 18, 2024.

**Table 2. Baseline characteristics of the patients with multiple system atrophy cerebellar variant (MSA-C) and the control patients.**

| | MSA-C | controls | p-value |
|---|---|---|---|
| age (year) | 60.7±14.1 | 48.2±9.9 | 0.16 |
| sex | 3/3 | 3/3 | 1.00 |
| disease | MSA-C | MS, NMO, SCA | n.a. |
| ataxia | 6/6 | 2/6 | 0.01* |
| autonomic dysfunction | 6/6 | 2/6 | 0.01* |
| parkinsonism | 2/6 | 1/6 | 0.54 |
| family history | 0/6 | 1/6 | 0.34 |
| disease duration (year) | 2.7±1.0 | 4.7±4.2 | 0.28 |

Ataxia and autonomic dysfunction were more frequently observed in MSA-C patients. No other significant differences were observed in age, sex, and disease duration.

MSA-C: multiple system atrophy cerebellar variant, MS: multiple sclerosis, NMO: neuromyelitis optica, SCA: spinocerebellar atrophy, n.a.: not assessed.

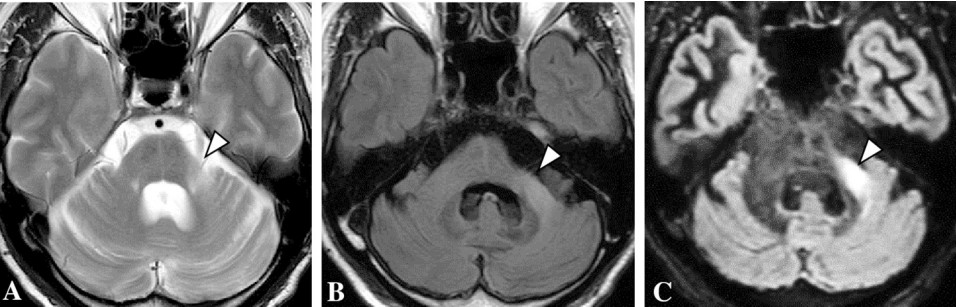

**Fig 1. Examples of bright middle cerebellar peduncle (MCP) signs in T2-weighted image (T2WI), fluid-attenuated inversion recovery (FLAIR), and double inversion recovery (DIR) sequences.** (A) T2WI. (B) FLAIR. In the (C) DIR sequence, bright MCP sign (arrowhead) is more clearly visible. Data from these first imaging of case 9 are presented in Table 1.

## Clinical evaluation

Diagnoses were verified by two board-certified neurologists (W.S. and T.H., with 13 and 17 years of experience, respectively) according to established Gilman's MSA diagnostic criteria [1]. All patients were examined within one month before the brain MRI. Clinical data including age, sex, disease duration, age at onset, parkinsonism, ataxia, and autonomic dysfunction were gathered from medical records.

## Image acquiring protocol

Brain MRI was performed on a 3.0T scanner (Ingenia 3.0T CX, PHILIPS Medical Systems Nederland B.V.). The scan parameters for DIR were as follows: field of view, 240 x 240 mm; section thickness: 1.6 mm; repetition time, 5500 ms; echo time, 266.87 ms, long inversion time, 2550 s, short inversion time 450 ms, and acquisition time 3 min 35 sec. In addition, we acquired FLAIR images (TR 10,000 ms and TE 120 ms) and T2WI (TR 3525 ms and TE 85 ms). Representative T2WI, FLAIR, and DIR images of the brainstem in MSA-C patients are shown in Fig 1.

## Image analysis

MCP signals were assigned a 4-point visual grade, as follows with reference to previous reports [12]: Grade 0, no visualization of bright MCP sign; Grade 1, a relatively bright MCP hyperintensity; Grade 2, definite bright MCP sign on a single slice; Grade 3, prominent bright MCP signs on two or more sequential slices (Fig 2). Also, the hot cross bun sign was assigned a 3-point visual grade, following the previous report [13]; Grade 0, no change; Grade 1, a vertical high-intensity line; Grade 2, a cruciform high-intensity line. The bright MCP sign and hot cross bun sign were visually assessed by two independent, experienced neurologists (W.S. and T.H.). The evaluation was performed in a blinded manner. If discrepancies existed between the two readers, the average of the two grades was utilized. Also, regarding the MCP sign, the signal intensity was measured in a circular region of interest (average diameter of 10 mm) within the thalamus (Th) and MCP on axial T2WI, FLAIR, and DIR sequences. The measurements of each structure were taken and presented as the mean value of the brightness (Fig 3). Next, MCP/Th ratios were calculated. The higher value on one side was adopted as the MCP/Th ratio of the patient because laterality was reported to be the nature of MSA [14, 15], and actually, some of our cases showed only a unilateral bright MCP sign.

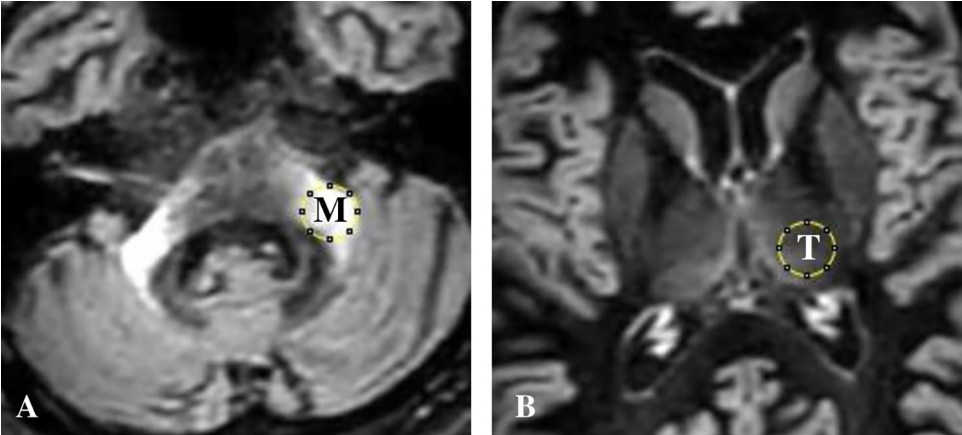

**Fig 2. Examples of the grades of bright middle cerebellar peduncle (MCP) sign.** (A) Grade 0, no visualization of bright MCP sign (arrows, control patient); (B) Grade 1, a subtle MCP hyperintensity (arrow); (C) Grade 2, a definite bright MCP sign on a single slice (arrow); (D) Grade 3, prominent bright MCP signs on two or more sequential slices (arrows, three sequential slices). All images are obtained by using a double inversion recovery procedure.

## Statistical analysis

Categorical variables were expressed as frequencies and percentages. Continuous variables were expressed as mean ± standard deviation. As appropriate, baseline characteristics were compared using either the chi-square test for categorical variables or the Mann–Whitney $U$ test for continuous variables. The level of statistical significance was set at $p < 0.05$. All statistical analyses were conducted using JMP Pro 12 software (SAS Institute, Cary, NC). Graphical images were built using PRISM 9 software (GraphPad Software, CA).

## Results

An example of bright MCP sign in an MSA-C patient is shown in Fig 1. In this case 9, the bright MCP sign is more clearly seen in the DIR than in the T2WI or FLAIR sequence.

Baseline characteristics of the MSA-C patients and control patients are shown in Tables 1 and 2. Among all the characteristics, only autonomic dysfunction and ataxia were statistically predominant in MSA-C patients. Also, although there was no statistical significance, the patients with MSA-C tended to be older and have a shorter medical history. There was no statistically significant correlation between disease duration and MCP signal grading or MCP/Th

**Fig 3. The signal intensity ratio of the middle cerebellar peduncle (MCP) lesion compared to the thalamus.** One centimeter diameter circular region of interest (ROI) is established on the MCP lesion (A), and an ROI of the same size is established on the ipsilateral MCP. M: MCP, T: thalamus. Both images are obtained by using a double inversion recovery procedure.

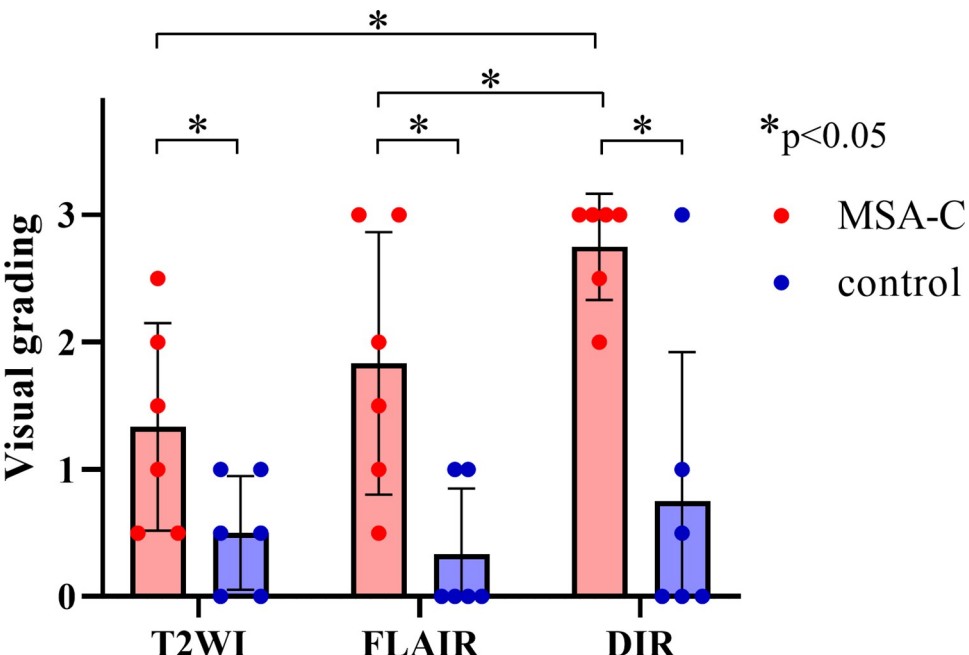

**Fig 4. Grade distribution of the bright middle cerebellar peduncle (MCP) sign in T2-weighted image (T2WI), fluid-attenuated inversion recovery (FLAIR), and double inversion recovery (DIR) sequences in patients with multiple system atrophy cerebellar variant (MSA-C) and control patients.** In all protocols, patients with MSA-C had significantly brighter MCP signs than control patients. DIR detected significantly brighter MCP signs than either T2WI or FLAIR. * p < 0.05.

ratio. Regarding laterality, the patient of case 9 had a bright MCP sign only on the left side and showed predominant left ataxia. The patient of case 11, however, had a bright MCP sign only on the left side, but showed no obvious laterality in ataxia (Table 1). All other MSA-C patients showed bilateral ataxia and bilateral bright MCP signs.

We evaluated T2WI, FLAIR, and DIR images for each patient and control. The κ value for interrater variability between the two examiners was 0.714 for the visual evaluation of the bright MCP sign. The average grades of the bright MCP sign in each group are shown in Fig 4. Among all the grades, the grade evaluated on the DIR sequence in the MSA-C group was higher. In three MSA patients (50%), the sign on the DIR image received a higher grade than the sign on the T2WI or FLAIR image. In the other 50% of cases, signs on the DIR and T2WI or FLAIR were given equivalent grades; in no case was the grade lower for the sign on DIR than for the sign on T2WI or FLAIR. The statistical analyses revealed that DIR is more a sensitive detector of the bright MCP sign than T2WI or FLAIR. We also measured the MCP-thalamus signal intensity ratio (MCP/Th ratio). A comparison of MCP/Th ratios measured by each of the imaging protocols in MSA-C cases and controls revealed that only DIR was able to distinguish MSA-C cases with statistically significant differences (p = 0.032) (Fig 5). This result shows that it was significantly easier to visualize the bright MCP sign on DIR than on T2WI or FLAIR. Regarding the hot cross bun sign, grades from DIR images were significantly higher than grades from FLAIR images but not statistically significantly different from grades from T2WI images (Fig 6). In addition, in two cases where MRI scans were performed over time (case 9 and case 11 in Table 1), the brightness of MCP increased and appeared in the contralateral lesion (Fig 7). In case 9, only a unilateral bright MCP sign was present at onset, but one year later bright MCP signs appeared bilaterally. The bright MCP signs were most evident in the DIR sequence (Fig 7A-7F). In case 11, the bright MCP sign became clearer every six

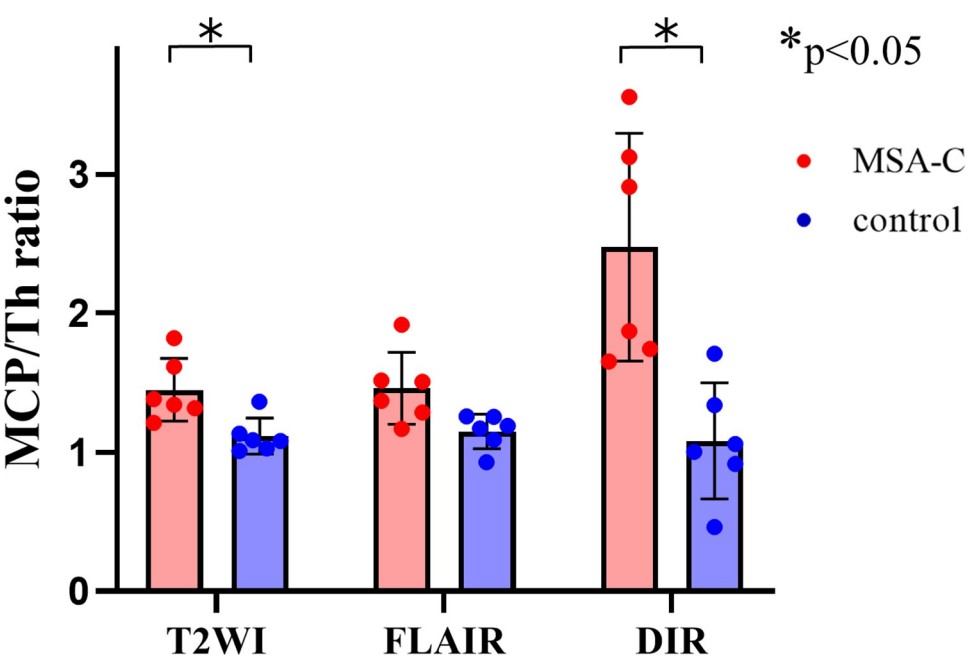

**Fig 5. Middle cerebellar peduncle (MCP)/thalamic ratio of signal intensity.** The MCP/thalamic ratios were highest in double inversion recovery sequences in patients with multiple system atrophy cerebellar variant, which was statistically significantly different from MCP/thalamic ratios in control patients. * p < 0.05.

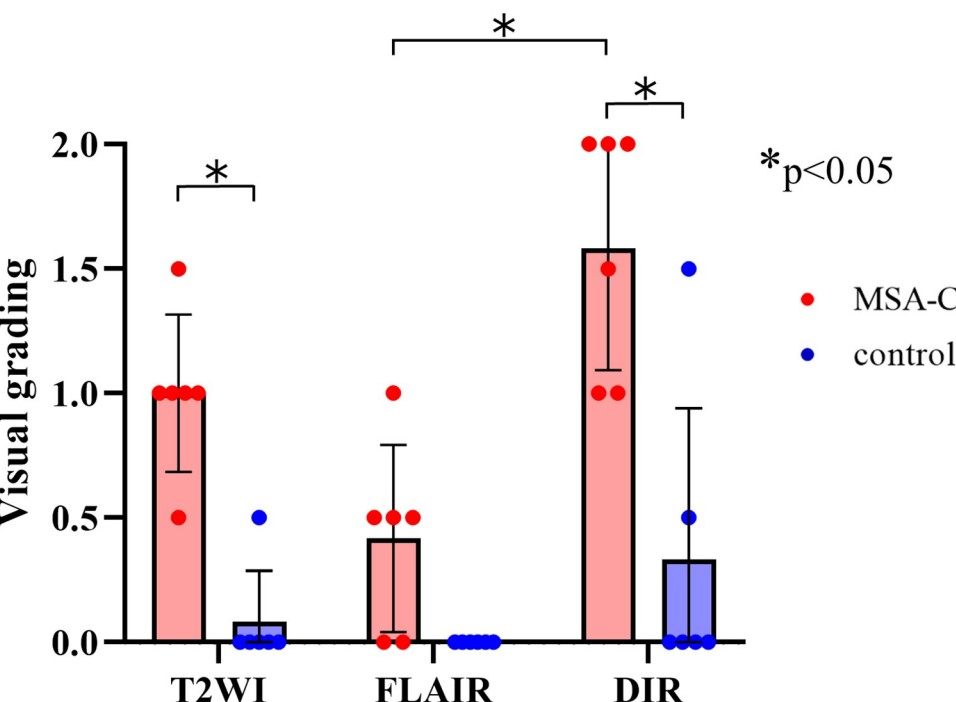

**Fig 6. Hot cross bun sign grading in T2-weighted image (T2WI), fluid-attenuated inversion recovery (FLAIR), and double inversion recovery (DIR) sequences.** Using a visual 3-point grading system, we showed that hot cross bun signs on T2WI and DIR sequences were significantly more obvious in the multiple system atrophy cerebellar variant group than the control group. Detection of the hot cross bun sign was significantly greater by DIR than by FLAIR, but not significantly different between DIR and T2WI. * p < 0.05.

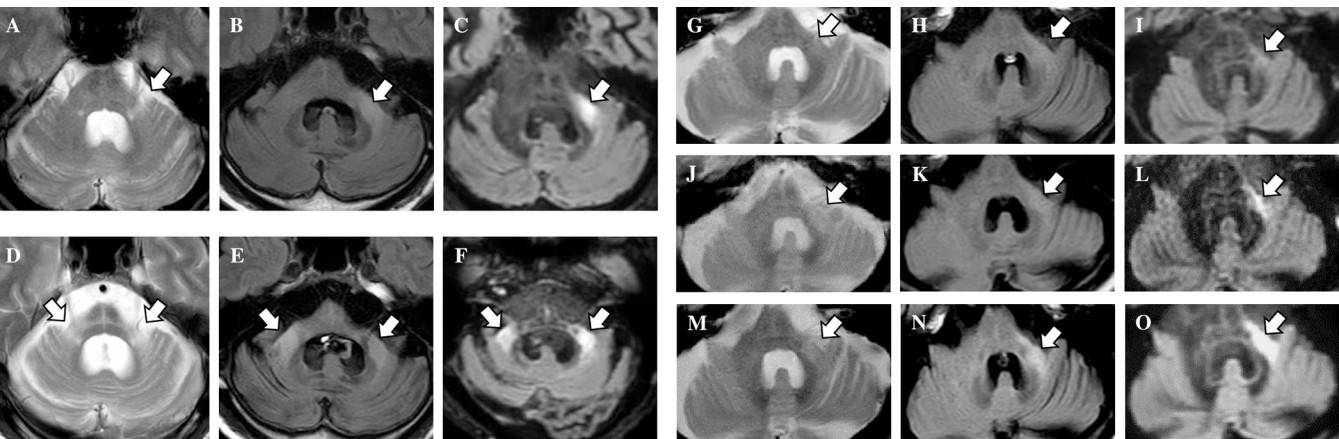

**Fig 7. Middle cerebellar peduncle (MCP) signals in multiple system atrophy cerebellar variant patients over time.** The bright MCP sign increased in intensity over time and appeared on the contralateral side in cases that could be followed over time. In case 9, only a unilateral bright MCP sign was present at disease onset (A-C), but one year later, bright MCP signs appeared bilaterally (D-F). The bright MCP signs were most evident in double inversion recovery sequences. In case 11, the bright MCP sign was subtle at the time of the diagnosis (G-I), became clearer six months later (J-L), and was prominent at one year after disease onset (Figure M-O). A, D, G, J, and M are T2-weighted images. B, E, H, K, and N are fluid-attenuated inversion recovery images. C, F, I, L, and O are double inversion recovery images.

months and was prominent at one year after onset (Fig 7G-7O). In cases 9 and 11, the brightness of the MCP sign on T2 and FLAIR images increased as time passed. Subjectively, however, the MCP sign appeared to have the highest signal intensity on DIR. However, due to the small number of cases, statistical analysis was not possible. In case 9, ataxic symptoms were initially limited to one-half of the body and expanded bilaterally 10 months later. The MRI taken 12 months later showed bilateral expansion of the bright MCP sign. Accordingly, in case 9, the symptoms and images were concordant, but in case 11, the lesions were unilateral on the image, but the symptoms were bilateral from the time of the first visit. More cases are needed to compare imaging findings and clinical symptoms.

## Discussion

The results of our study show that the DIR protocol is more sensitive for revealing the bright MCP sign in MSA-C patients than conventional protocols (T2WI and FLAIR). Previous studies about bright MCP signs in MSA-C used T2WI, FLAIR [4], or T1/T2 ratio [6]; however, as far as we know, no previous reports have focused on bright MCP signs on DIR images. And the usefulness of DIR in detecting bright MCP signs in patients with MSA-C is still unrevealed.

The main pathological feature of MSA is the widespread presence of GCIs in the central nervous system, and the density of GCIs that contain α-synuclein is reported to be significantly correlated with disease progression [16]. The pontocerebellar fibers in the MCP have been shown to be one of the main regions where GCI pathology progresses in an early phase of the MSA disease [17]. The accumulation of GCIs is also reported to be correlated with myelin loss [17], meaning myelin fibers of MCP are affected in the early stage of MCA-C. On the other hand, the DIR sequence was first reported in 1994 by Redpath et al. [18] to give a more precise delineation of the cerebral cortex. DIR provides two inversion pulses that attenuate signals from CSF and white matter to achieve superior delineation between gray matter and white matter [18]. So, we hypothesized why DIR is superior to T2WI or FLAIR for detecting bright MCP signs as follows. DIR was reported to show a better rate of infratentorial lesion abnormality detection than FLAIR or T2WI [11] because DIR suppresses CSF and white matter signals.

Brainstem lesions, which are white matter-rich and surrounded by CSF, may be appropriate targets of DIR imaging. By conventional MRI methods, the bright MCP sign is affected by CSF signals in T2WI and white matter signals in FLAIR. DIR was expected to suppress CSF and white matter signals, making the MCP signal more prominent and allowing the MCP signal to be detected at an earlier stage.

Our study has several limitations. First, statistical insufficiency is possible since this study consists of a small number of cases at a single institution. Secondly, all of the cases in our study were clinically but not pathologically diagnosed and thus were not cases of definitely diagnosed MSA. It has been reported that among cases of MSA-C diagnosed before death, pathologically confirmed MSA accounts for 62% to 92% [19]. Third, this was a cross-sectional study, and we did not investigate the course (timeline) of disease progression in all the patients, so a prospective study is needed. Fourth, our control groups were not healthy individuals and included patients with cerebellar involvement, which may cause pseudo-positivity.

## Conclusion

Compared to T2WI or FLAIR, DIR is potentially a more sensitive method of detecting the bright MCP sign in MSA-C, and the addition of DIR should be considered in imaging studies of MSA cases. The hot cross bun sign tended to be more identifiable in DIR images than in conventional MRI images but the difference was without statistical significance. In addition, the MCP abnormality increased over time. In cases not initially diagnosed, imaging follow-up over time is desirable.

## Acknowledgments

We thank all patients who took part in this study and their families. We thank Johshin Matsu-zaki, Yasunori Saho, and all the other radiological technicians at Kokura Memorial Hospital for their excellent work.

## Author Contributions

**Conceptualization:** Wataru Shiraishi, Ayano Matsuyoshi, Yusuke Nakazawa, Yukiko Inamori, Akira Yogi, Tetsuya Hashimoto.

**Data curation:** Wataru Shiraishi, Ayano Matsuyoshi, Yusuke Nakazawa, Yukiko Inamori, Tetsuya Hashimoto.

**Formal analysis:** Wataru Shiraishi, Akira Yogi, Tetsuya Hashimoto.

**Funding acquisition:** Wataru Shiraishi.

**Investigation:** Wataru Shiraishi, Ayano Matsuyoshi, Tetsuya Hashimoto.

**Methodology:** Wataru Shiraishi, Akira Yogi, Tetsuya Hashimoto.

**Project administration:** Wataru Shiraishi.

**Resources:** Wataru Shiraishi.

**Software:** Wataru Shiraishi.

**Supervision:** Akira Yogi, Tetsuya Hashimoto.

**Validation:** Wataru Shiraishi, Tetsuya Hashimoto.

**Visualization:** Wataru Shiraishi.

**Writing – original draft:** Wataru Shiraishi.

**Writing – review & editing:** Wataru Shiraishi, Ayano Matsuyoshi, Yusuke Nakazawa, Yukiko Inamori, Akira Yogi, Tetsuya Hashimoto.

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
