## [Decision Letter · Decision Letter 0]

12 Sep 2024

PONE-D-24-27948Bright middle cerebellar peduncle sign in multiple system atrophy with predominant cerebellar ataxia is more apparent in double-inversion recovery than in conventional imagingPLOS ONE

Dear Dr. Shiraishi,

Thank you for submitting your manuscript to PLOS ONE. After careful consideration, we feel that it has merit but does not fully meet PLOS ONE’s publication criteria as it currently stands. Therefore, we invite you to submit a revised version of the manuscript that addresses the points raised during the review process.

We look forward to receiving your revised manuscript.

Kind regards,

Kensaku Kasuga

Academic Editor

PLOS ONE

Journal Requirements:

2. In the online submission form, you indicated that information on this paper can be obtained by contacting the corresponding author only for the parts of the information that do not conflict with private information. 

Additional Editor Comments:

Please revise your manuscript according to the reviewers' suggestions and provide a point-by-point response to the reviews.

Reviewers' comments:

Reviewer's Responses to Questions

**Comments to the Author**

1. Is the manuscript technically sound, and do the data support the conclusions?

Reviewer #1: Yes

Reviewer #2: Partly

Reviewer #3: Yes

2. Has the statistical analysis been performed appropriately and rigorously? 

Reviewer #1: Yes

Reviewer #2: Yes

Reviewer #3: Yes

3. Have the authors made all data underlying the findings in their manuscript fully available?

Reviewer #1: Yes

Reviewer #2: Yes

Reviewer #3: Yes

4. Is the manuscript presented in an intelligible fashion and written in standard English?

Reviewer #1: No

Reviewer #2: Yes

Reviewer #3: No

5. Review Comments to the Author

Reviewer #1: This study reports that DIR is the most useful MRI sequence for evaluating the MCP sign in MSA-C compared to other MRI sequences. The results are reasonable, however, several additional investigations are necessary to enhance it as an original research article.

Major Points:

・The study focuses only on the MCP sign, without evaluating other findings. The imaging diagnosis of MSA-C should be comprehensive. It is worth considering whether the Hot Cross Bun sign is also better visualized with DIR.

・MSA-C is a relatively common condition. It would be preferable to evaluate the utility of DIR in a larger cohort, including cases that are difficult to diagnose in the early stages. The control group in such a study should include patients with Parkinson’s disease and other atypical parkinsonism.

Minor Points:

・Native English editing is recommended.

・Some figures are not correctly aligned with their descriptions.

・For Figure 2, provide continuous slices for Grade 3.

・As this is a retrospective study, the baseline characteristics of the MSA-C patients and controls should be included in the Methods and Subjects section.

・There is some redundancy between the Discussion and Introduction sections.

Reviewer #2: The authors have shown that DIR is useful for visual assessment of the Bright MCP sign. Although the small number of cases studied limits statistical interpretation, the usefulness of DIR is clearly demonstrated.

However, there are some concerns that need to be addressed as listed below.

1. The authors need to explain why they chose multiple sclerosis, neuromyelitis optica, and spinocerebellar atrophy as control groups.

2. In the text and figure legends, 'T2' should be written as 'T2WI' for T2-weighted images.

3. I think the corresponding figure is different from the instructions referring to the figure in the text. Are Figures 2, 3, and 1 in the text really Figures 1, 2, and 3, respectively? Also, I believe that the A, B, B, C listed in the figure legend in Figure 2 are correctly A, B, C, D. Manuscripts must be carefully and meticulously prepared.

Reviewer #3: Comments to the Author

It has been a pleasure to review your manuscript entitled “Bright middle cerebellar peduncle sign in multiple system atrophy with predominant cerebellar ataxia is more apparent in double-inversion recovery than in conventional imaging.”

I find the study to be worthy of publication; however, several concerns must be addressed before it can be considered for final acceptance. Please refer to the following comments:

Major comments:

1. “This study aimed to investigate whether DIR can detect the bright MCP sign in MSA-C patients significantly better than T2 or FLAIR. Furthermore, in cases where we could follow the MRI changes over time, we evaluated the time-course change of the MCP abnormality. ” (P4, line 66-68)

Given the stated aim, would it not be sufficient to analyze only the MSA-C group? Please provide a clear and detailed explanation of the reasons for including control subjects and the results derived from this comparison.

Moreover, as the authors mention in the limitations, the control group consisting of a mixture of demyelinating diseases, astrocytopathy, and degenerative diseases seems inappropriate. Table 1 shows a comparison between the MSA-C group and this control group, but even if significant differences are observed, it is difficult to assess the relevance of these findings when comparing such a mixed group. I recommend reevaluating the control selection.

2. Please include a summary of the MSA-C cases in the manuscript. Information such as initial symptoms, duration of illness, and the timing of MRI scans are crucial to understanding the context of the findings.

3. It appears that Figures 2 and 3 may be reversed. Please carefully review and confirm their correctness.

4. In Figure 6, the authors show the progression of MCP signals in MSA-C patients. When specifically does the MCP signal increase in MSA-C? Is there a noticeable difference between cases with unilateral versus bilateral MCP signals? I recommend discussing and interpreting the results of Figure 6 in greater detail.

Minor comments:

1. The figure legends are lacking detail. For instance, all figures should indicate the MRI sequence used. Additionally, Figure 1D is not sufficiently explained. Please review and enhance the figure legends accordingly.

2. Certain parts of the text are underexplained. For example, is the case presented in Figure 1 included in your overall analysis? Additionally, which specific cases correspond to case 1 (P8, line 141) and Case 2 (P8, line 143)? Please clarify these points with more detail.

3. There is inconsistency in the use of parentheses around figure references, such as Figure 1 (P7, line 122) and (Figure 5) (P8, line 141). Please ensure consistent formatting throughout the manuscript.

4. Proper capitalization needs to be applied throughout the text. For instance, "case" (P8, line 141), "Case" (P8, line 143), "middle" (P16, line 301), and "double" (P16, line 302) should all be capitalized where appropriate. Please carefully proofread the manuscript to correct these and other minor errors.

6. PLOS authors have the option to publish the peer review history of their article (what does this mean?). If published, this will include your full peer review and any attached files.

Reviewer #1: No

Reviewer #2: No

Reviewer #3: **Yes: **AKIO AKAGI

---

## [Author Response · Author response to Decision Letter 0]

4 Oct 2024

In the manuscript, the responses to the comments of reviewer 1 are highlighted in green, responses to the comments of reviewer 2 are highlighted in blue, and responses to the comments of reviewer 3 are highlighted in yellow. The responses to the comments of two or more reviewers are highlighted in gray.

Reviewer #1: This study reports that DIR is the most useful MRI sequence for evaluating the MCP sign in MSA-C compared to other MRI sequences. The results are reasonable, however, several additional investigations are necessary to enhance it as an original research article.

Major Points:

・The study focuses only on the MCP sign, without evaluating other findings. The imaging diagnosis of MSA-C should be comprehensive. It is worth considering whether the Hot Cross Bun sign is also better visualized with DIR.

Thank you for pointing this out. We also evaluated the hot cross bun sign. However, it wasn't easy to assess quantitatively. The hot cross bun sign cannot be evaluated quantitatively like the MCP/Th ratio, so only a subjective grade was assigned. We have added the results of that (Figure 6). Regarding the hot cross bun sign, the grade on DIR was statistically significantly higher than the grade on FLAIR, but not higher than the grade on T2WI (P2-3, L35-37, P7, L117-120, P9, L162-164, P12, L214-216, Fig 6, P19, L358-361).

・MSA-C is a relatively common condition. It would be preferable to evaluate the utility of DIR in a larger cohort, including cases that are difficult to diagnose in the early stages. The control group in such a study should include patients with Parkinson’s disease and other atypical parkinsonism.

Thank you for the suggestion. We also wished to have other diseases such as PSP, CBS, Parkinson's disease, etc. However, the DIR procedure is added based on the physician's decision and not included in the basic MRI protocol. Therefore, most of the controls were patients with demyelinating disease. We stated the reason in the “Materials and methods” section (P5, L81-88).

Minor Points:

・Native English editing is recommended.

Thank you for your suggestion. An English editing company edited our manuscript. We have attached the certification.

・Some figures are not correctly aligned with their descriptions.

Thank you for pointing this out. We have checked and revised the alignment of the figures with their descriptions.

・For Figure 2, provide continuous slices for Grade 3.

Thank you for your suggestion. We modified Figure 2D.

・As this is a retrospective study, the baseline characteristics of the MSA-C patients and controls should be included in the Methods and Subjects section.

Thank you for the suggestion. We made a new table of baseline characteristics of patients with MSA-C and control patients and cited the new table in the Materials and methods section (Table 1, P5, L81-88).

・There is some redundancy between the Discussion and Introduction sections.

Thank you very much. We have edited and shortened the two sections (P4, L67, P12, L200).

Reviewer #2: The authors have shown that DIR is useful for visual assessment of the Bright MCP sign. Although the small number of cases studied limits statistical interpretation, the usefulness of DIR is clearly demonstrated.

However, there are some concerns that need to be addressed as listed below.

1. The authors need to explain why they chose multiple sclerosis, neuromyelitis optica, and spinocerebellar atrophy as control groups.

Thank you for the suggestion. At our hospital, the physician decides whether to perform the DIR procedure, which is not part of the basic MRI set. Therefore, most of the control patients were diagnosed with demyelinating diseases. We added the sentences to the Materials and methods section (P5, L81-88).

2. In the text and figure legends, 'T2' should be written as 'T2WI' for T2-weighted images.

Thank you for pointing this out. We changed “T2” to “T2WI.”

3. I think the corresponding figure is different from the instructions referring to the figure in the text. Are Figures 2, 3, and 1 in the text really Figures 1, 2, and 3, respectively? Also, I believe that the A, B, B, C listed in the figure legend in Figure 2 are correctly A, B, C, D. Manuscripts must be carefully and meticulously prepared.

Thank you for pointing this out. We checked and revised these discrepancies.

Reviewer #3: Comments to the Author

It has been a pleasure to review your manuscript entitled “Bright middle cerebellar peduncle sign in multiple system atrophy with predominant cerebellar ataxia is more apparent in double-inversion recovery than in conventional imaging.”

I find the study to be worthy of publication; however, several concerns must be addressed before it can be considered for final acceptance. Please refer to the following comments:

Major comments:

1. “This study aimed to investigate whether DIR can detect the bright MCP sign in MSA-C patients significantly better than T2 or FLAIR. Furthermore, in cases where we could follow the MRI changes over time, we evaluated the time-course change of the MCP abnormality. ” (P4, line 66-68)

Given the stated aim, would it not be sufficient to analyze only the MSA-C group? Please provide a clear and detailed explanation of the reasons for including control subjects and the results derived from this comparison.

Moreover, as the authors mention in the limitations, the control group consisting of a mixture of demyelinating diseases, astrocytopathy, and degenerative diseases seems inappropriate. Table 1 shows a comparison between the MSA-C group and this control group, but even if significant differences are observed, it is difficult to assess the relevance of these findings when comparing such a mixed group. I recommend reevaluating the control selection.

Thank you for pointing this out. No previous reports have evaluated MCP sign with DIR, and it was not known how the MCP sign would appear in healthy subjects or patients with other diseases. Therefore, we used other diseases as a control group. We added this information to the manuscript. We also wanted to have a patient with typical Parkinson's disease in the control group. However, DIR is an optional procedure that is added based on the physician's decision and is not one of our hospital's basic MRI protocols. So, most of the control subjects had demyelinating diseases, the assessment of which is the most common reason for performing DIR. We added the reasoning in the “Materials and methods” section. Because of the retrospective nature of this study, we cannot add other control groups. If needed, we will start a new prospective study now, but it may take 3 to 4 more years to complete (P5, L81-88).

2. Please include a summary of the MSA-C cases in the manuscript. Information such as initial symptoms, duration of illness, and the timing of MRI scans are crucial to understanding the context of the findings.

Thank you for pointing it out. We added Table 1 listing the patients’ and control patients’ characteristics and added that information to the manuscript (P5, L81-85). 

3. It appears that Figures 2 and 3 may be reversed. Please carefully review and confirm their correctness.

Thank you for pointing this out. We checked and corrected the problem.

4. In Figure 6, the authors show the progression of MCP signals in MSA-C patients. When specifically does the MCP signal increase in MSA-C? Is there a noticeable difference between cases with unilateral versus bilateral MCP signals? I recommend discussing and interpreting the results of Figure 6 in greater detail.

Thank you for pointing this out. We added the information in Table 1. Because of the small number of patients, we could not assess the association between unilateral/bilateral MCP signs and symptoms (Table 1, P8, L144-149, P10, L170-178).

Minor comments:

1. The figure legends are lacking detail. For instance, all figures should indicate the MRI sequence used. Additionally, Figure 1D is not sufficiently explained. Please review and enhance the figure legends accordingly.

Thank you for pointing it out. We checked and added the information and MRI sequences (P18, L332, 338, 344).

2. Certain parts of the text are underexplained. For example, is the case presented in Figure 1 included in your overall analysis? Additionally, which specific cases correspond to case 1 (P8, line 141) and Case 2 (P8, line 143)? Please clarify these points with more detail.

Thank you. We changed the case presentation according to information provided in Table 1.

3. There is inconsistency in the use of parentheses around figure references, such as Figure 1 (P7, line 122) and (Figure 5) (P8, line 141). Please ensure consistent formatting throughout the manuscript.

Thank you very much. We changed the manuscript to ensure consistency of formatting.

4. Proper capitalization needs to be applied throughout the text. For instance, "case" (P8, line 141), "Case" (P8, line 143), "middle" (P16, line 301), and "double" (P16, line 302) should all be capitalized where appropriate. Please carefully proofread the manuscript to correct these and other minor errors.

Thank you for pointing it out. We checked and revised it.

---

## [Decision Letter · Decision Letter 1]

29 Oct 2024

Bright middle cerebellar peduncle sign in multiple system atrophy with predominant cerebellar ataxia is more apparent in double-inversion recovery imaging than in conventional imaging

PONE-D-24-27948R1

Dear Dr. Shiraishi,

We’re pleased to inform you that your manuscript has been judged scientifically suitable for publication and will be formally accepted for publication once it meets all outstanding technical requirements.

Kind regards,

Kensaku Kasuga

Academic Editor

PLOS ONE

Reviewers' comments:

Reviewer's Responses to Questions

**Comments to the Author**

1. If the authors have adequately addressed your comments raised in a previous round of review and you feel that this manuscript is now acceptable for publication, you may indicate that here to bypass the “Comments to the Author” section, enter your conflict of interest statement in the “Confidential to Editor” section, and submit your "Accept" recommendation.

Reviewer #2: All comments have been addressed

Reviewer #3: All comments have been addressed

2. Is the manuscript technically sound, and do the data support the conclusions?

Reviewer #2: Yes

Reviewer #3: Yes

3. Has the statistical analysis been performed appropriately and rigorously? 

Reviewer #2: Yes

Reviewer #3: Yes

4. Have the authors made all data underlying the findings in their manuscript fully available?

Reviewer #2: Yes

Reviewer #3: Yes

5. Is the manuscript presented in an intelligible fashion and written in standard English?

Reviewer #2: Yes

Reviewer #3: Yes

6. Review Comments to the Author

Reviewer #2: The authors have shown that DIR is useful for visual assessment of the Bright MCP sign. They have revised the manuscript appropriately in line with the previous comments.

Reviewer #3: Comments to the author:

It is my pleasure to have the opportunity to review this manuscript once again.

The authors have responded appropriately to the comments. I consider it worthy of publication.

I am praying for all of authors success in the future.

7. PLOS authors have the option to publish the peer review history of their article (what does this mean?). If published, this will include your full peer review and any attached files.

Reviewer #2: No

Reviewer #3: **Yes: **AKIO AKAGI

---

## [Editor Report · Acceptance letter]

31 Oct 2024

PONE-D-24-27948R1 

PLOS ONE

Dear Dr. Shiraishi, 

I'm pleased to inform you that your manuscript has been deemed suitable for publication in PLOS ONE. Congratulations! Your manuscript is now being handed over to our production team.

Kind regards, 

on behalf of

Dr. Kensaku Kasuga 

Academic Editor

PLOS ONE